# Benefits on Hematological and Biochemical Parameters of a High-Intensity Interval Training Program for a Half-Marathon in Recreational Middle-Aged Women Runners

**DOI:** 10.3390/ijerph19010498

**Published:** 2022-01-03

**Authors:** Jèssica B. Bonet, Casimiro Javierre, João Tiago Guimarães, Sandra Martins, David Rizo-Roca, Jorge Beleza, Ginés Viscor, Teresa Pagès, José Magalhães, Joan R. Torrella

**Affiliations:** 1Secció de Fisiologia, Departament de Biologia Cellular, Fisiologia i Immunologia, Facultat de Biologia, Universitat de Barcelona, 08028 Barcelona, Spain; jbonetbo8@alumnes.ub.edu (J.B.B.); david.rizo.roca@ki.se (D.R.-R.); jmbeleza@fade.up.pt (J.B.); gviscor@ub.edu (G.V.); tpages@ub.edu (T.P.); 2Departament de Ciències Fisiològiques, Facultat de Medicina i Ciències de la Salut, Campus de Bellvitge, Universitat de Barcelona, 08907 Barcelona, Spain; cjavierre@ub.edu; 3Serviço de Patologia Clínica, Centro Hospitalar São João, 4200-319 Porto, Portugal; jtguimar@med.up.pt (J.T.G.); u006810@chsj.min-saude.pt (S.M.); 4LaMetEx–Laboratory of Metabolism and Exercise, Faculdade de Desporto, Centro de Investigação em Atividade Física e Lazer (CIAFEL), Universidade do Porto, 4200-450 Porto, Portugal; jmaga@fade.up.pt

**Keywords:** high intensity interval training, moderate intensity continuous training, recreational running, muscle damage, inflammatory, oxidative stress

## Abstract

(1) Background: half-marathon races are popular among recreational runners, with increases in participation among middle-aged and women. We aimed to determine the effects of two half-marathon training programs on hematological and biochemical markers in middle-aged female recreational runners; (2) Methods: ten women (40 ± 7 years) followed moderate intensity continuous training (MICT), based on running volume below 80% $\dot{V}$O_2_max, and another ten women followed high intensity interval training (HIIT) at 80%–100% $\dot{V}$O_2_max, with less volume, and combined with eccentric loading exercise. Hematology, plasma osmolality, and plasma markers of metabolic status, muscle damage, inflammatory, and oxidative stress were measured before (S1) and after (S2) training and 24 h after the half-marathon (S3); (3) Results: both training programs had similar moderate effects at S2. However, the acute response at S3 induced different alterations. There was a greater decrease in cholesterol and triglyceride levels in MICT and reductions in markers of damage and inflammation in HIIT. Greater variability in some plasma markers at S3 in MICT suggests that there is inter-individual variability in the response to training; (4) Conclusions: HIIT led to better adaptation to the competition maybe because of the repeated exposure to higher oxygen consumption and eccentric loading exercise.

## 1. Introduction

Although half-marathons are not included in the Olympic Games, they are extremely popular among recreational runners, as demonstrated by increased participation over recent decades [1]. Although still present, the gender gap in participation has narrowed considerably and, according to the latest Running USA report [2], in 2019 female half-marathon finishers equaled male’s numbers. Knechtle et al. [1] found that the mean age of half-marathon finishers in Switzerland between 1999 and 2014 was approximately 41 years for both sexes. The rapid growth in race participation by women has had important performance consequences and has been the result of the development of specific training programs. Regarding athletic performance, we must consider several established physiological differences between men and women, including erythrocyte mass, body composition, and maximum oxygen consumption ($\dot{V}$O_2_max) [3]. For example, the linear relationship between hemoglobin mass and $\dot{V}$O_2_max in humans [4] could account for the differences in $\dot{V}$O_2_max values reported between elite females (67.1 ± 4.2 mL·kg^−1^·min^−1^) and elite males (74.1 ± 2.6 mL·kg^−1^·min^−1^) marathon runners in USA [5]. However, the loss of maximal strength and the rate of force development after a half-marathon were found to be similar between men and women, these force capacity reductions being associated with central and peripheral fatigue in both sexes [6]. Interestingly, some performance responses appear to favor female recreational runners, as it is the case of the reported differences in the running strategy and the control of fatigue: women are better in managing rhythm control during the second half of a marathon [7]. There are also known differences between women and men regarding metabolism during long-term aerobic effort. For example, at the same relative intensity toward the end of the exercise, women oxidized 47% more plasma-free fatty acids than men [8]. Women also show less dependence on liver and muscle glycogen for endurance effort, relying more on intramyocellular lipids, which is attributed to their higher estrogen levels [9]. Research among women aged 25–35 years revealed that after a nine week training (three times a week) there was a 7.6% decrease in total cholesterol, 8% increase in high-density lipoprotein (HDL), and no change in low-density lipoprotein (LDL), hemoglobin, serum glucose, or triglycerides (TAG) [10].

Over the last decade, researchers have evaluated different plasma markers and hematological parameters after a half-marathon run, with most focused on young males. Plasma inflammatory and immune response markers [11] have been reported to increase in an exercise dose-dependent manner. Markers of cardiac injuries, such as troponin, creatine kinase, and myoglobin, also seem to increase transiently in men after half-marathons [12], with no differences between sexes when women were also included [13]. By contrast, Abbasi et al. [14] reported sex-specific differences in inflammation-related pathway activation and leukocyte gene regulation in response to acute exhaustive exercise. Lippi and coworkers [15,16] also reported the kinetics of the variation from the baseline (3–24 h) of several plasma markers of muscle damage [15] and hematological parameters [16] after a half-marathon in a mixed-age male sample. Although they included females in a more recent experimental design, they did not analyze the male and female data separately [17].

As evidenced, there is now significant interest concerning the biochemical and physiological changes associated with running a half-marathon race. However, to our knowledge, few studies have tracked these changes before and after the half-marathon training period and compared the results to the post-race recovery period. We aimed to fill this gap by analyzing the long-term response to different 12-week training programs for a half-marathon among premenopausal amateur female runners: a moderate intensity continuous training (MICT) or a high intensity intermittent training (HIIT) program. We have chosen this demographic group because it has recently experimented with one of the greatest increases in half-marathon participation both in Europe [1] and in the USA [2]. Moreover, women aged 35–45 years frequently experience serious difficulties reconciling daily professional and family life with amateur running. Training programs aimed to reduce running volume (HIIT) in favor of intensity could be a valuable and attractive training alternative to training programs based on longer running times (MICT) [18].

We were interested in analyzing the possible differences in the hematological indicators and the muscle damage, inflammation, and oxidative stress markers after the two training programs. The rationale for testing such markers was to assess if the adaptations induced by training were different depending on the training program. Moreover, since hemolysis, osmotic shock, muscle damage, or oxidative stress could derive from activities with a greater peripheral impact, we would like to ascertain if an acute high impact activity, such as a half-marathon, could provoke different responses depending on the training protocol followed. Finally, analyzing masters runners following a high intensity training program has an additional interest since it has recently been found that high intensity sports provide higher levels of anti-inflammatory cytokines than low and moderate intensity sports [19] and that an elevated anti-inflammatory response to acute exercise may be an adaptive response to lifelong training [20]. Hypothetical differences between training groups, especially after performing acute high impact exercise, would suggest that a particular training protocol would result in a more protective phenotype.

## 2. Materials and Methods

### 2.1. Participants

Data from this work derived from a greater study [18] aimed to analyze several functional and performance parameters after following MICT or HIIT training programs for a half-marathon. Because of this, the sample size was powered on the performance variable “finishing time” in the half-marathon. To fit power parameters of α = 0.05 and β = 0.20, estimating both the size of the change to be detected and the size of the standard deviation change as 0.05, the minimum sample size was set at *n* = 10. Thus, twenty healthy female master recreational runners were chosen for this study after being recruited from different running clubs in Barcelona (Spain). The volunteers aged 40 ± 7 years (mean ± SD), with 61 ± 7 kg of body mass, a height of 167 ± 6 cm, and a body mass index of 22.5 ± 2.8 (kg·m^−2^). They were required to be regular runners with minimum training routines of 5 h over 3 days/week, have previous experience running half-marathons, be premenopausal, be non-smokers, have no injuries, and take no medications (including oral contraceptives). They had the mean following performance characteristics: $\dot{V}$O_2_max of 48.0 ± 5.6 mL·kg^−1^·min^−1^; basal heart rate of 78 ± 13 bpm; and a best half-marathon finishing time of 01:59:36 (h:min:sec). Participants were randomly assigned to two training groups of ten runners each. One group followed the MICT program, with a high-volume and low-intensity training protocol. The other group followed a HIIT program, completing a low-volume and high intensity interval protocol that included bodyweight resistance exercises. At the beginning of the intervention period, there were no significant differences between groups in either performance or anthropometric parameters. Since the menstrual cycle may affect parameters derivative from blood samples, the participants recorded the duration of their menstrual cycles (t, in days) and monitored their basal temperatures (T, in °C) during the menses days. No statistical differences were found neither in the duration; nor in basal temperature between the women assigned to MICT (t = 4.6 ± 1.1, T = 36.1 ± 0.37) or HIIT (t = 5.2 ± 1.4, T = 36.1 ± 0.46).

### 2.2. Training Programs

After 1 month of rest in August, participants started training in September and finished in December after completing three non-consecutive training sessions every week for 12 weeks. The details of each training program are published elsewhere [18]. Briefly, the MICT program was based on that reported by Galloway and Galloway [21]. It consisted of 2 days of continuous running for 40 min and 60 min, alternated with 1 day of long-distance running (12–25 km, with two sessions where the whole half-marathon distance was run in 7th and 9th weeks) and 800 m intervals each week. The HIIT program was designed to reduce training volume, increase training intensity, and add eccentric loading via jump training and downhill running [18]. HIIT training comprised the following each week: a session of long-distance running (8–16 km, with the maximum distance run in the 6th week), a session of interval running (200 m and 400 m in series), and a session of uphill and downhill (for recovery) was alternated with fast running and combined with a 12-station circuit of bodyweight resistance exercises performed at maximum intensity. To improve anaerobic power and increase the efficiency of using elastic energy, the participants performed 50 m sprints at the end of the third session. HIIT participants invested 33 h and 26 min, a 17% reduction in time compared with the MICT group (40 h and 30 min). Women in the HIIT group ran 301 km, which was 21% less than that run by women in the MICT group (383 km). After the training programs, all women participated in the same half-marathon race held in Vilanova i la Geltrú (Spain), located at sea level on the Mediterranean coast (41°13′27″ N, 1°43′33″ E).

### 2.3. Blood Sampling and Preparations

All venous blood samples were taken by conventional clinical procedures from the antecubital vein and using EDTA as an anticoagulant. Blood samples (S) from each participant were collected at three different times: S1, at 1 week before training began; S2, at 48 h before the half-marathon; and S3, at 24 h after finishing the race. Thus, S1 will be the baseline data, S2 will show the effect of the different training programs, and S3 will inform on the half-marathon impact on the hematological and plasma parameters. The blood collection was done at the same time of the day (6.30 AM) for each sample point. Participants were asked to fast for 8 h and refrain from drinking coffee or any other stimulating substance 24 h prior to sampling. An aliquot of fresh blood was separated for hematological determination. White blood cells (WBC), red blood cells (RBC), platelets, hemoglobin concentration, mean corpuscular volume (MCV), mean corpuscular hemoglobin (MCH), and mean corpuscular hemoglobin concentration (MCHC) were quantified in a hematological analyzer (Celltac α MEK-6318K, Nihon Kohden Europe GmbH, Rosbach, Germany). The remaining blood material, which was approximately 10 mL, was immediately centrifuged at 3000× *g* rpm for 10 min to separate out the plasma. An aliquot was used to measure plasma osmolality in a micro osmometer (Model 3300, Advanced Instruments Inc., Norwood, MA, USA) and the remaining plasma aliquots were rapidly frozen and stored at −80 °C for later biochemical analysis.

### 2.4. Biochemical Assays

All samples were analyzed in triplicate, and the mean of three values was used for statistical analysis. Plasma ion determination was done in an electrolyte analyzer (ISElyte-X9, Tecil, Barcelona, Spain). Protein content was assayed spectrophotometrically using bovine serum albumin as the standard, according to Lowry’s method.

Commercial assay kits from Elabscience Biotechnology Inc (BioNova, Madrid, Spain) were used for plasma albumin, glucose, urea, aspartate aminotransferases (AST and ALT), and gamma-glutamyl transferase (GGT) determination. Commercial colorimetric kits, ab235627 and ab196994 from Abcam Inc. (Abcam Inc., Cambridge, MA, USA), were used to measure plasma bilirubin and aldolase, respectively. Plasma creatine kinase (CK) activity was determined spectrophotometrically using a commercial test kit (Horiba-ABX A11A01632, Montpellier, France). C-reactive protein (CRP) was measured using an enzyme-linked immune sorbent assay system ELISA-PENTRA 400 (Horiba ABX, Montpellier, France).

Total antioxidant status (TAS) was measured spectrophotometrically using a commercial kit (Randox Laboratories Ltd NX2332, Crumlin, UK). Superoxide dismutase (SOD) activity was measured spectrophotometrically at 550 nm using a commercial Ransod kit from Randox (catalog no. SD 125, Crumlin, UK). The activity of glutathione peroxidase was assayed spectrophotometrically at 340 nm using a commercial kit from Randox Laboratories Ltd (NX2332, Crumlin, UK). The activity of glutathione reductase was measured with a spectrophotometric procedure at 340 nm using a commercial kit from Randox Laboratories Ltd (NX2332, Crumlin, UK). A medium containing perchloric acid at 5% (*w*/*v*) was used to precipitate the proteins from aliquots for glutathione assay. To measure TGSH content, samples were centrifuged for 1 min at 13,000× *g* after neutralization with potassium hydrogen carbonate (0.76 M), and a supernatant aliquot was incubated for 15 min at 30 °C in a microtiter plate with a reagent solution containing NADPH (1.68 mM) and 5,5′dithio-bis (2-nitrobenzoic acid) (0.7 mM). A kinetic analysis was performed at 412 nm after the addition of 20 U·mL^−1^ of glutathione reductase. GSSG content was measured by adding 2-vinylpyridine, giving a final concentration of 5% (*v*/*v*), before neutralization to inactivate the sulfhydryl groups. The assay used the same steps for TGSH measurement. Both TGSH and GSSG concentrations were calculated based on calibration curves made with commercial standards. Reduced glutathione (GSH) was calculated as GSH = TGSH – GSSG, while the percent oxidized glutathione (%GSSG) was calculated as the percentage of the ratio GSSG/TGSH.

### 2.5. Statistical Analysis

After checking for normality (Kolmogorov–Smirnov test) and homoscedasticity (Levene test), data were analyzed by one-way ANOVA with repeated measures to compare the values obtained at each sampling time (i.e., S1, S2, and S3) within each training program. Post hoc multiple comparisons after ANOVA testing were performed using the Holm-Sidak method. *p*-values were considered significantly different when *p* < 0.05. Data are reported as means ± standard deviations in the text and tables, unless otherwise stated. In the box-and-whisker plots, boxes represent the interquartile range (first and third) separated by the median, the black dot is the mean, and the whisker ends are the minimum and maximum values. Several linear correlation analyses were run using the finishing time of the half-marathon (in seconds), as a dependent variable, and muscle damage, inflammation, and most relevant oxidative stress markers after the race (S3), as independent variables. Pearson product-moment correlation coefficients and *p*-values for the linear correlations were calculated. Cohen’s d values were calculated in the comparisons between S3 vs. S2 and S3 vs. S1 for muscle damage, inflammation, and relevant oxidative stress markers. Their values are given in the figure and table legends with ANOVA *p*-values. All data were statistically analyzed using Sigma Plot v. 11 (Systat Software, Inc., San José, CA, USA).

## 3. Results

There were no significant statistical differences between groups at baseline (S1) in any of the variables studied. Since the objective of the study was to compare the effect of training and the impact of a half-marathon run on hematology and several plasma markers within a MICT and a HIIT program, no between-group comparisons were performed. Thus, comparisons between S1, S2, and S3 are referred to within groups (MICT or HIIT). Both training programs had similar effects on all analyzed parameters after 12 weeks of training (S1 versus S2), except for hematocrit, interleukin-6 (IL-6), and TAS. However, the acute responses after the half-marathon (S3) revealed differences in blood markers of metabolic status, muscle damage, inflammation, and oxidative stress. The results of the linear correlations using the performance of the race (i.e., finishing times) as dependent variable and the muscle damage, inflammation, and most relevant oxidative stress markers as independent variables, did not show significant linear correlations for any variable, as is deduced from Pearson product-moment correlation coefficients (r) and *p*-values (Table 1).

### 3.1. Hematological Parameters

Direct measures of the RBC count and hemoglobin concentration did not show statistically significant differences after the training programs (S2) or the half-marathon race (S3) (Figure 1A,B). However, compared with S1, hematocrit levels in the MICT group increased significantly by S2 and decreased significantly by S3 (Figure 1C). These changes in the hematocrit were not observed in the HIIT group, where values remained unchanged. Derived measures (Table 2) from red blood cell count and hemoglobin concentration (mean corpuscular concentration, MCH), and hemoglobin concentration and hematocrit (mean corpuscular hemoglobin concentration, MCHC) differed significantly from S1 to S2 in the MICT group (*p* < 0.001), but not in the HIIT group. However, the half-marathon run elicited significant decreases in MCH and MCHC in both training groups. The MCV only showed a higher statistically significant (*p* < 0.05) value between S3 and S1 in the HIIT group. At S3, WBC counts increased in both groups, but this was only significant in the HIIT group; by contrast, the platelet count decreased significantly in both groups (Table 2).

### 3.2. Plasma Solute Concentrations

Figure 2 shows the total plasma osmolality and plasma solute concentrations. Both training programs (S2) significantly reduced plasma osmolality (*p* < 0.01), and samples obtained 24 h after the half-marathon (S3) showed significant decreases compared with pre-race (S2; *p* < 0.05) and pre-training (S1; *p* < 0.001) values (Figure 2A). Plasma sodium and chloride concentrations increased in the MICT group between S2 and S3 (*p* = 0.092 for sodium and *p* < 0.05 for chloride), but they showed no significant changes in the HIIT group (Figure 2B,C). Conversely, plasma phosphate concentrations decreased between S2 and S3: this was significant in the MICT group (*p* < 0.01) and marginal in the HIIT group (*p* = 0.081). Moreover, in both groups, samples obtained at S3 had significantly lower plasma phosphate values than those obtained at S1 (Figure 2D). Box plots showing glycaemia in S3 indicate a wide range of plasma glucose concentrations with means tending to decrease in both training groups (Figure 2E). This decrease was significant (*p* < 0.05) when considering the post hoc S1 vs. S3 comparison in the MICT group. Urea plasma concentrations decreased gradually, but significantly, from S1 to S3 in the MICT group, while no statistically significant differences were observed between any sample points in the HIIT group (Figure 2F).

### 3.3. Liver and Lipid panel

Although there was a trend for values to decrease from S1 to S2, the training modality had no significant effect on any parameter related to liver or lipid panels, as deduced from the absence of statistical differences after both training programs (Table 3). Moreover, the half-marathon run (S3) elicited no alterations in plasma liver enzymes (AST, ALT, and GGT) or total plasma bilirubin. However, comparing results at S3 results with those at S1 and S2, starkly significant decreases were evident in total plasma protein, albumin content, and much of the lipid panel. It was notable that the most striking differences between the S3 and the S1/S2 values occurred in the MICT group. Decreases in plasma total protein and albumin concentration ranged from 24% to 30% in the MICT group, contrasting with the 11% to 17% decreases in the HIIT group. Similarly, total cholesterol and LDL decreased from 30% to 45% in the MICT group compared with 16% to 24% in the HIIT group. Finally, TAG plasma levels were significantly lower in the MICT group at S3 than at S1 (*p* < 0.001) and S2 (*p* < 0.01), a finding that contrasted with the lack of a statistically significant difference in the HIIT group.

### 3.4. Muscle Damage and Inflammatory Markers

Neither training program had an effect on plasma CK levels (i.e., no significant difference from S1 to S2). However, significant elevations in this marker of muscle damage were evident in both groups at S3: the post hoc multiple comparison tests indicated a significant difference between S3 and S1/S2 (Figure 3A). The mean increase in CK from S2 to S3 differed significantly between the training groups, being more pronounced with MICT (153% increase) than HIIT (63% increase) (*p* = 0.043, not shown in Figure 2A). Aldolase levels did not present significant differences between sampling times in either training group (Figure 3B). The pattern for CRP was similar to that for CK, with no significant differences between S1 and S2, but with significantly elevated values at S3 that were more pronounced in the MICT group (Figure 3C) than in the HIIT group (*p* = 0.039; not shown in Figure 2C). IL-6 levels increased significantly from S1 to S3 in both groups, and while there was no difference between S1 and S2 in the MICT group, there was a markedly significant increase from S1 to S2 in the HIIT group (*p* < 0.001; Figure 3D). Finally, it is notable that the MICT group showed greater variability in S3 values than the HIIT group for all markers of muscle damage and inflammation, as shown by the higher interquartile ranges in the box-and-whisker plots in Figure 2.

### 3.5. Oxidative Stress-Related Parameters

TAS showed a trend to increase after MICT and had a significantly higher value after HIIT (*p* < 0.05), but these reverted to baseline after the half-marathon (Table 4). Among the antioxidant enzymes, no statistically significant differences were found between the three sampling times, except for a significant increase in SOD activity at S3 in the MICT group. From S1 to S2, neither MICT nor HIIT significantly affected the levels of either TGSH or GSH; however, both parameters increased markedly by S3 (*p* < 0.01 and *p* < 0.001, respectively). There was a pronounced decrease in GSSG after both training programs, but this was only statistically significant in the HIIT group (*p* < 0.01). Decreases in the %GSSG were significant (*p* < 0.05) after both exercise programs. Notably, the half-marathon race provoked marked and significant increases in TGSH, GSH, and GSSG. Regarding the %GSSG, there was a non-significant increase from S1 to S3 in both groups. The wide range of %GSSG values are noteworthy, as deduced from the high standard deviation figures.

## 4. Discussion

In a previous study [18] we evaluated and compared different functional and performance parameters of the two training programs. Both programs were effective in improving performance with mean finishing time reductions in the half-marathon (min:sec) ranging from 2:29 (HIIT) to 3:50 (MICT). Here we present the hematological and biochemical results. These results contribute to increasing our knowledge of the hematological and biochemical alterations in the plasma of female recreational athletes after the acute bout of endurance exercise. Several studies in the past decade have assessed the changes in hematological parameters, muscle damage, oxidative stress, and inflammatory plasma markers elicited by a 21 km run in populations with different fitness levels and ages, but mainly among men [11,12,13,15,16,17,22,23,24,25,26]. The relevance and novelty of our study is that we analyzed hematological parameters and biochemical plasma markers in a poorly studied group, namely middle-aged women training at a recreational level. Moreover, we evaluated the effects of two 12-week training programs, with different intensities and running volumes, on blood plasma levels of various hematological and biochemical parameters. This experimental approach allows for the comparison of not only the hematological and biochemical changes taking place 24 h after an acute endurance exercise prepared for with different training strategies but also the effects of those strategies on the analyzed markers.

### 4.1. Hematological Parameters

Given the absence of statistically significant differences in the RBC count and hemoglobin concentration after both training programs compared to baseline (Figure 1A,B), the significant increase in hematocrit after MICT was a surprise (Figure 1C). The greater relative increase in erythrocyte volume (MCV) in this group, which matched the significant decreases in hematimetric indices related to corpuscular hemoglobin (Table 2), could explain this difference. A significant decrease in MCH and MCHC after MICT (not observed in HIIT) may indicate a greater erythrocyte volume that supports the significant increase in hematocrit associated with the more aerobic training schedule. The hematocrit elevation following MICT reverted to baseline values 24 h after the race (Figure 1C), resulting in a significant reduction from S2 to S3 that was not observed in the HIIT group. Several hypotheses could be proposed for the post-race decrease in hematocrit following MICT. These include exercise-induced hemolysis derived from foot strikes during running [27] and oxidative damage or perturbed osmotic homeostasis of erythrocytes [28]. However, the absence of statistical differences in RBC counts or hemoglobin concentrations between the pre- and post-race values suggest that this hypothesis should be viewed with caution. A more likely explanation for the decreased hematocrit in the MICT group after the half-marathon is that plasma volume expansion occurred during recovery from the acute endurance exercise, consistent with previous findings [29,30]. Moreover, it has been reported that the intensity and duration of exercise is the major stimulus for plasma volume expansion [31].

In both training programs, there was a significant decrease in both the MCH and MCHC at 24 h after the half-marathon run compared to baseline (Table 2), which is compatible with the greater erythrocyte volume found after the race (significant for HIIT group). These results support those of previous half-marathon studies in men and women of similar ages and in male sub-elite athletes that revealed increases in the RBC distribution width 20 h after a race [17], together with 7% increases in reticulocyte mean volume 48 h after competition [22]. As an acute strenuous exercise, the half-marathon provides a strong stimulus for reticulocyte release, and may increase the variation in RBC volume and size.

The training protocols did not affect the WBC count (i.e., S2 vs. S1), but higher values were evident after the half-marathon (S3), reaching significance for the HIIT group. Similar increases were observed in other studies of runners [22,32,33]. The increase in WBC after prolonged exercise may indicate neutrophil marginalization or inflammation secondary to tissue destruction, as suggested elsewhere. Duca et al. [22] found a 62% increase in the number of polymorphonuclear neutrophils just after finishing a half-marathon, which reverted to normal by 48 h. After high intensity effort, increases in the circulating WBC count have been observed for up to 12 h, with differences in leukocytosis depending on the terrain in which the exercise is performed [33]. Given that our results show a greater and significant increase in the HIIT group, we conclude that the type of training affects the magnitude of response, being greater with greater intense and eccentric loads.

No differences were found in platelet counts due to the training programs, but significant decreases were evident in both groups after the half-marathon (Table 2). Similar findings were found 24 h after finishing a marathon in adolescents [34], while Lippi et al. [17] reported no changes by 20 h after a half-marathon. However, Lippi et al. [35] investigated the acute effects of a half-marathon on various blood coagulation parameters in a group of 33 middle-aged males and found increased prothrombotic activity 3 h after exercise. Regarding the specific effect on platelets, discrepancies exist between different studies depending on the training degree, previous pathology, and the effort type [36]. It was noteworthy that, after 24 h, a prolonged and strenuous effort through a half-marathon was a sufficiently powerful stimulus to induce a 14% reduction in platelet counts compared with the pre-race values, regardless of the training followed (S2 to S3). This response could be the consequence of an adrenergic stimulus from the final phase of the race, generating prothrombotic conditions that affect the platelet phase of coagulation, inducing greater and more effective adhesion and a certain pro-activation environment of the platelet mass. This could facilitate the repair of the injured endothelium, triggering a local response that increases oxygen supply after platelet activation.

### 4.2. Plasma Solute Concentrations

Exercise stimulates antidiuretic hormone and aldosterone secretion, even during the effort itself. This compensates for the water lost by sweating and the filtration in active tissues due to vasodilation produced by the metabolites released during muscular activity. The net effect would be an increase in plasma volume, which could explain the significant reduction in osmolality after both training programs (Figure 2A). In fact, hypervolemia is a hallmark of endurance training that is manifested by elevated plasma volume and RBC corpuscular volume (up to 10%) [37]. This adaptation has recently been described in aerobic efforts after intermittent activities [38], and it could be accompanied by an increase in performance [39,40].

After 12 weeks, no significant differences were found between the training programs in the major components of plasma osmolality (i.e., sodium, chloride, glucose, and urea; Figure 2B–F). However, there were non-significant reductions (*p* > 0.05) in plasma total protein and albumin levels in the MICT group (5.2%–6.8%) and the HIIT group (4.6%) that could partially account for the decreased osmolality in both groups (Table 3). Following the half-marathon, we observed a different response between the MICT and HIIT groups. Women in the MICT group showed a two-fold percent greater decrease in plasma osmolality, total protein, and albumin from S2 to S3 compared with the HIIT group (Figure 2A, Table 3), as well as greater significant decreases in plasma phosphate, glucose, and urea concentrations (Figure 2D–F). Changes in albumin have been related to acute inflammatory processes in certain pathologies [41], and in our study, appeared to have been caused by the acute stress produced by the half-marathon, leading to increased vascular permeability in response to tissue injury. Under this premise, those in the MICT group tolerated acute exercise less well than those in the HIIT group. The significant increase in plasma chloride concentrations in the MICT group, contrasting with the absence of changes in the HIIT group (Figure 2C), further supports this finding. Indeed, it has been shown that the total and extracellular water, sodium, and chloride content of muscle increases after any injury that promotes an inflammatory response [42].

The decrease in plasma osmolality 24 after the half-marathon in middle-aged recreational female runners in both training groups was consistent with the 3.8% significant increase in plasma volume reported by Lippi et al. [15] for middle-aged recreational male runners. Although it has been reported after extremely severe and prolonged exercise sessions [43], particularly in overhydrated conditions where there is excessive fluid intake, we found no signs of exercise-associated hyponatremia. In fact, the plasma osmolality was still low 24 h after the race, while both the chloride and sodium concentrations in the plasma were higher in the HIIT group than in the MICT group. Thus, the more extensive training regimen followed by the MICT group predisposed the middle-aged women in our study to have a greater capacity to cope with the stress of sweating-related water and micronutrient loss when racing and recovering [44].

### 4.3. Liver and Lipid Panel

Although the significant reduction in plasma albumin concentration after the half-marathon could suggest disturbed liver cell integrity, we found no statistically significant differences in any other liver biochemical markers (Table 3). As such, we conclude that neither MICT nor HIIT was injurious to liver cells, at least for up to 24 h after finishing the half-marathon. It also seems reasonable to exclude the hypothetical possibility of hemolysis caused by structural changes in the red cell membranes [45], given that bilirubin levels did not change after either training or the half-marathon race. Regarding the hepatic markers, Niemelä et al. [23] studied six recreational runners (four men and two women) of varying ages and reported that AST and ALT values remained slightly higher 48 h after the race, but that GGT remained unchanged. Lippi et al. [15] reported statistically significant differences in AST after 24 h in a group of 15 middle-aged men. The differences in the characteristics of the studied populations could explain some of the contrasting results in these previous investigations.

The type of training (MICT or HIIT) had no effect on plasma lipid markers in the present study, with no significant change from S1 to S2 (Table 3). The long-term maximum effort elicited by the half-marathon did promote significant reductions (*p* < 0.001 in all cases) after the race (S3) when compared to pre-race values (S2) in circulating markers of lipid metabolism, albeit with magnitudes that varied with the training program. Total cholesterol was reduced by 30% with MICT and 16% with HIIT, while LDL decreased by 36% with MICT and 24% with HIIT. More consistent differences were found in plasma HDL and TAG levels, which showed statistically significant decreases (*p* < 0.05) of 10% in HDL and 40% in TAG (*p* < 0.001) with MICT; this contrasted with a lack of significant changes (*p* = 0.165) with HIIT. These results indicate that the intensity of training resulted in different lipid profiles after the race. This finding supports the conclusions of research in middle-distance and marathon runners [46], which suggested that the effect of an acute bout of maximal endurance exercise on lipid and lipoprotein parameters may have been related to different training histories.

### 4.4. Muscle Damage and Inflammation Markers

Supporting data previously published on the acute variation of muscle damage and inflammatory biochemical markers after a half-marathon [11,15,23,24,25,26], we showed that middle-aged women engaged in both training groups exhibited signs of muscle damage and inflammatory response after the half-marathon (Figure 3). In fact, despite starting from comparable post-training conditions at S2 compared to baseline at S1, the physiological impact of the run caused increases in CK, CRP, and IL-6 by S3 compatible with a pattern of exercise-induced muscle damage described elsewhere [47,48]. It is interesting to note that the results obtained for muscle damage and inflammation (and for oxidative stress) markers after the race (S3) were not conditioned by the performance taken during the competition (assessed after the finishing times obtained by the runners). This conclusion is derived from values of the linear correlations (r and *p*) presented in Table 1: there were not significant (*p* > 0.05) linear correlations between performance and any analyzed parameter, with low Pearson coefficients (Table 1).

The delayed kinetics of plasma CK appearance after exercise-induced muscle damage indicate that biochemical analysis performed later during recovery, at 48 h or 72 h, may evidence higher CK levels. In the present research, it was notable that the half-marathon had a greater impact among women in the MICT group than in the HIIT group. A *t*-test for two markers of damage and inflammation between the MICT and HIIT groups at S3 revealed significantly higher values in the MICT group for CK (*p* = 0.043) and CRP (*p* = 0.039) (Figure 3A,C, *p*-values not shown). One possible explanation for these results could derive from the different maximum distances run between MICT and HIIT participants during their training sessions. Since the HIIT program was designed to reduce volume and increase the intensity, women engaged in this program never run during their training the whole half-marathon distance (with a maximum of 16 km in the 6th week). In contrast, women following the MICT program run half-marathon distances in two sessions (in the 7th and 9th weeks) [18]. Moreover, the reported differences in muscle damage and inflammation markers at S3 between MICT and HIIT could also be explained after the greater intensity training followed introduced in HIIT. The addition of downhill running and jump training sessions in the HIIT group would have enhanced the tolerance of the musculature for repeated loading promoting a better skeletal muscle adaptation to this physiological assault. The increase in IL-6 levels at S2 confirms the effect of eccentric loading exercise in the HIIT group. Regarding muscle damage and inflammatory response, skeletal muscle can adapt to systematic strength training sessions to alleviate the impact of a subsequent harmful stimuli, a phenomenon known as the repeated bout effect [49]. Thus, considering the oscillating trajectory of a half-marathon, with the need for continuous random variation, the adoption of strength training in HIIT likely promoted a repeated bout effect that protected participants to a greater extent than through MICT, particularly due to the inclusion of eccentric training.

Figure 3 also shows greater variability (higher interquartile ranges) in the MICT group for all plasma markers of muscle damage and inflammation at S3. This finding indicates a difference in the acute response to exercise depending on the intensity and volume of training. Moreover, it is in accordance with the inter-individual variability found in some performance and physiological indicators in response to these training programs [50].

### 4.5. Markers of Oxidative Stress

Data from the present study highlight that, despite their different approaches, both training regimens and the half-marathon modulated the redox environment in participants. In fact, as evidenced in Table 4, participants from both the MICT and HIIT groups showed increased total antioxidant capability based on their TAS, the increase in GSH levels, and the decrease in %GSSG levels. Of note, the modulation toward a more reductive environment was more evident in the HIIT group than in the MICT group (e.g., TAS and %GSSG values), probably reflecting the specific features of each regimen. Unfortunately, there is a lack of data to support the distinct impact of each training method from a mechanistic perspective on oxidative stress.

It is also relevant that even 24 h after the half-marathon, GSH levels remained elevated compared to pre-race (S3 vs. S2). This finding contrasts with the return to baseline values by 24 h for other indicators of oxidative stress [24]. Our results suggest that racing induced a state of increased oxidative stress that was maintained for at least 24 h in both groups. Glutathione levels were evaluated in plasma samples and significantly enhanced oxidative stress at the tissue level eventually caused GSH exportation from the liver to the muscle to cope with the fall in skeletal muscle antioxidant capacity. Given that oxidative stress-related parameters were also evaluated 24 h after the race, it is also possible that some of the levels relate to muscle damage, as previously described. In fact, increased production of oxygen reactive species in the setting of enhanced oxidative stress are considered the key mechanistic steps in the pathophysiology of exercise-induced muscle damage [51]. The excessive release of pro-inflammatory cytokines, such as IL-6 (Figure 3D), can increase the production of reactive oxygen species and trigger oxidative stress in some tissues [52].

Our study presents some limitations that must be acknowledged. The primary limitation was the small sample size of the training groups due to the difficulty in finding participants that fit the population group intended to study. The effect of low sample size was quantified calculating Cohen’s d values in inflammation, oxidative stress, and muscle damage parameters, showing in some variables a marked effect size that must be taken into consideration when interpreting the results. The limited available information on the nutritional data of the participants during the training programs and before the half-marathon run must be also acknowledged as a potential confounder factor. Additionally, variability in the premenopausal status of the women may have also played a role in the dispersion found in some parameters. However, despite these limitations, we believe that our results reveal specific-training effects on hematological and biochemical parameters in middle-aged recreational runners after training and running a half marathon.

## 5. Conclusions

Excluding the hematocrit, IL-6, and TAS, both training programs had similar moderate effects on all parameters after 12 weeks of training in premenopausal women following different training programs for a half-marathon. However, the acute response to the half-marathon, measured after 24 h, induced a temporary state of fatigue-related alterations in recovery and stress reflected in blood markers of muscle damage, inflammation, oxidative stress, and metabolic status. It was notable that differences existed depending on the training adopted: the incorporation of eccentric loads via jump training and downhill running, the running volume, and its duration. HIIT that incorporated strength exercises based on body weight led to better adaptation to competition because of the repeated exposure to higher oxygen consumption and anaerobic metabolism during the intense bouts of exercise. This training, therefore, provides an advantage, giving a more homogeneous inter-individual response before the onset of muscle damage and inflammation.

## Figures and Tables

**Figure 1 ijerph-19-00498-f001:**
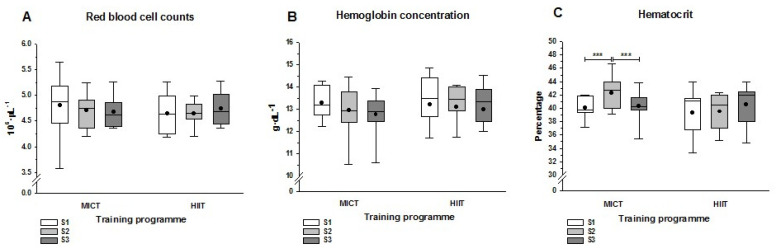
Hematological parameters at three sampling points. (**A**) Red blood cell counts; (**B**) Hemoglobin concentration; (**C**) Hematocrit at three sampling points. Sampling was before (S1) and after (S2) the MICT or HIIT programs, as well as 24 h after the half-marathon race (S3). The three asterisks show the statistically significant differences (*p* < 0.001) between each pair of sampling time points. The lines and dots inside the boxes represent the median and the mean, respectively, and the whisker ends the minimum and maximum values. Abbreviations: HIIT, high intensity interval training; MICT, moderate intensity continuous training.

**Figure 2 ijerph-19-00498-f002:**
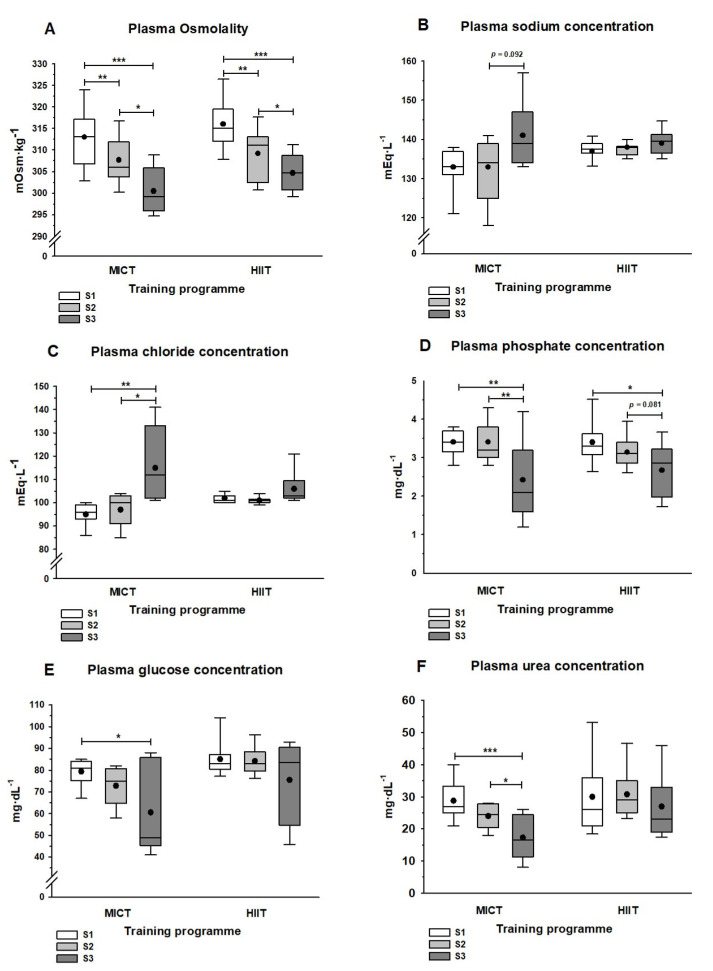
Plasma osmolality-related measurements at three sampling points. Sampling was before (S1) and after (S2) the MICT or HIIT programs, as well as 24 h after the half-marathon race (S3). (**A**) Measured osmolality; (**B**) Sodium; (**C**) Chloride; (**D**) Phosphate; (**E**) Glucose; (**F**) Urea. Asterisks show the statistically significant differences between each pair of sampling time points: *, *p* < 0.05; **, *p* < 0.01; and ***, *p* < 0.001. When almost significant differences were obtained, the *p*-value is indicated. The lines and dots inside the boxes represent the median and the mean, respectively, and the whisker ends the minimum and maximum values. Abbreviations: HIIT, high intensity interval training; MICT, moderate intensity continuous training.

**Figure 3 ijerph-19-00498-f003:**
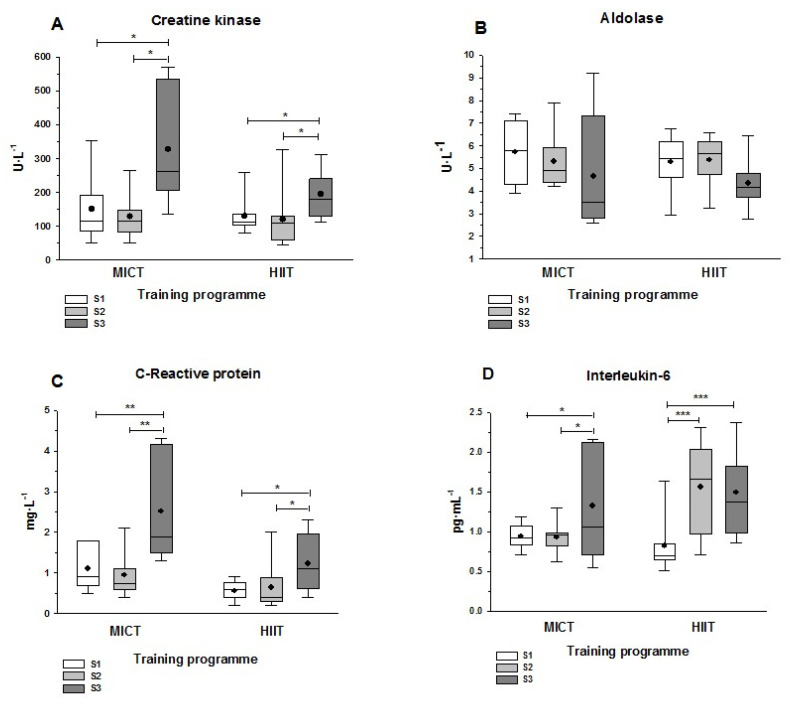
Plasma muscle damage and inflammation markers at three sampling points. Sampling was before (S1) and after (S2) the MICT or HIIT programs, as well as 24 h after the half-marathon race (S3). (**A**) Creatine kinase; (**B**) Aldolase; (**C**) C-Protein reactive; (**D**) Interleukin-6. Asterisks show the statistically significant differences between each pair of sampling time points: *, *p* < 0.05; **, *p* < 0.01; and ***, *p* < 0.001. Cohen’s d values in the comparisons between two groups were as follows: (**A**) MICT: d = 1.04 (S1 vs. S2), d = 1.05 (S2 vs. S3); HIIT: d = 0.94 (S1 vs. S2), d = 0.75 (S2 vs. S3). (**B**) MICT: d = 0.51 (S1 vs. S2), d = 0.54 (S2 vs. S3); HIIT: d = 0.78 (S1 vs. S2), d = 0.78 (S2 vs. S3). (**C**) MICT: d = 1.14 (S1 vs. S2), d = 1.09 (S2 vs. S3); HIIT: d = 1.07 (S1 vs. S2), d = 0.93 (S2 vs. S3). (**D**) MICT: d = 0.75 (S1 vs. S2), d = 0.73 (S2 vs. S3); HIIT: d = 1.21 (S1 vs. S2), d = 1.28 (S2 vs. S3). The lines and dots inside the boxes represent the median and the mean, respectively, and the whisker ends the minimum and maximum values. Abbreviations: HIIT, high intensity interval training; MICT, moderate intensity continuous training.

**Table 1 ijerph-19-00498-t001:** Results of the linear correlation analyses using the finishing time of the half-marathon (in seconds) as dependent variable and markers after the race (S3) as independent variables.

	MICT (*n* = 10)		HIIT (*n* = 10)	
	r	*p*	r	*p*
Creatine kinase	−0.567	0.240	−0.263	0.434
Aldolase	−0.533	0.173	0.439	0.177
C-Reactive protein	0.047	0.929	−0.635	0.091
Interleukin-6	−0.045	0.896	−0.263	0.434
TAS	−0.264	0.567	0.099	0.773
SOD	0.426	0.341	−0.128	0.724
TGSH	−0.571	0.236	−0.431	0.394

SOD, superoxide dismutase; TAS, total antioxidant status; TGSH, total glutathione.

**Table 2 ijerph-19-00498-t002:** Hematimetric indices and formed elements count values (white blood cells and platelets) of the high-volume and low-intensity (MICT) and low-volume and high-intensity training (HIIT) groups measured 1 week before training (S1), 48 h before the half-marathon(S2), and 24 h after the race (S3).

	MICT (*n* = 10)			HIIT (*n* = 10)		
Hematimetric Indices	S1	S2	S3	S1	S2	S3
MCV (fL)	85.6 ± 6.1	86.2 ± 5.9	86.4 ± 5.4	84.8 ± 3.6	85.1 ± 3.3	85.5 ± 3.4 ^*^
MCH (pg)	28.4 ± 2.3	27.2 ± 2.5 ^***^	27.4 ± 2.2 ^***^	28.5 ± 1.6	28.2 ± 1.7	27.2 ± 1.6 ^***###^
MCHC (g·dL^−1^)	33.2 ± 0.8	31.5 ± 0.9 ^***^	31.7 ± 1.1 ^***^	33.8 ± 0.9	33.2 ± 1.3	31.9 ± 0.8 ^***###^
**Formed elements (10^3^·µL^−1^)**						
WBC	5.95 ± 1.16	5.83 ± 1.12	6.10 ± 0.96	5.92 ± 1.45	5.43 ± 1.07	6.82 ± 1.54 ^#^
PLT	218 ± 45	218 ± 61	190 ± 36 ^*#^	209 ± 53	217 ± 42	181 ± 40 ^**##^

Statistically significant differences shown as asterisk (*) (vs. S1) and hags tag (^#^) (vs. S2): one symbol (*p* < 0.05), two symbols (*p* < 0.01), and three symbols (*p* < 0.001). Values are means ± SD. Abbreviations: MCHC, mean corpuscular hemoglobin concentration; MCH, mean corpuscular hemoglobin; MCV, mean corpuscular volume; PLT, platelets; WBC, white blood cells.

**Table 3 ijerph-19-00498-t003:** Liver and lipid panel values of the high-volume and low-intensity (MICT) and low-volume and high-intensity training (HIIT) groups measured 1 week before training (S1), 48 h before the half-marathon(S2), and 24 h after the race (S3).

	MICT (*n* = 10)			HIIT (*n* = 10)		
Liver Panel	S1	S2	S3	S1	S2	S3
Total protein (g·L^−1^)	67.2 ± 7.8	63.7 ± 8.5	48.6 ± 17.4 ^***##^	72.0 ± 4.2	68.7 ± 4.3	60.2 ± 15.3 ^*^
Albumin (g·L^−1^)	40.9 ± 3.5	38.1 ± 4.9	28.8 ± 9.6 ^***##^	42.2 ± 3.1	40.0 ± 2.6	35.6 ± 9.5 ^*^
Bilirubin (mg·L^−1^)	0.59 ± 0.15	0.47 ± 0.28	0.58 ± 0.41	0.49 ± 0.22	0.45 ± 0.13	0.56 ± 0.27
AST (U·L^−1^)	23.7 ± 5.2	24.4 ± 4.6	24.8 ± 11.0	22.1 ± 3.7	22.4 ± 4.4	23.5 ± 4.6
ALT (U·L^−1^)	12.9 ± 7.7	12.4 ± 3.6	12.7 ± 6.7	11.1 ± 2.5	10.3 ± 4.6	9.8 ± 2.4
GGT (U·L^−1^)	14.9 ± 5.4	17.1 ± 8.2	12.4 ± 7.9	12.9 ± 3.6	13.4 ± 4.5	12.0 ± 3.3
**Lipid panel (mg·L^−1^)**						
Total cholesterol	193 ± 34	187 ± 32	131 ± 33 ^***##^	188 ± 25	187 ± 32	157 ± 33 ^***##^
HDL	63.3 ± 8.8	57.9 ± 9.5	51.9 ± 18.9 ^*^	61.9 ± 11.6	63.2 ± 9.3	61.9 ± 15.5
LDL	118 ± 29	113 ± 28	72 ± 14 ^***##^	113 ± 25	114 ± 27	86 ± 21 ^***###^
TAG	62.0 ± 17.2	74.1 ± 16.7	44.3 ± 13.0 ^***##^	66.7 ± 20.2	58.5 ± 18.5	54.1 ± 15.4

Statistically significant differences shown as asterisk (*) (vs. S1) and hags tag (^#^) (vs. S2): one symbol (*p* < 0.05), two symbols (*p* < 0.01), and three symbols (*p* < 0.001). Values are means ± SD. Abbreviations: ALT, alanine aminotransferase; AST, aspartate aminotransferase; GGT, gamma-glutamyl transferase mean corpuscular volume; HDL, high-density lipoprotein; LDL, low-density lipoprotein; TAG, triglycerides.

**Table 4 ijerph-19-00498-t004:** Oxidative stress-related parameters of the high-volume and low-intensity (MICT) and low-volume and high-intensity training (HIIT) groups measured 1 week before training (S1), 48 h before the half-marathon(S2), and 24 h after the race (S3).

	MICT (*n* = 10)			HIIT (*n* = 10)		
	S1	S2	S3	S1	S2	S3
TAS (mmol·L^−1^)	1.29 ± 0.11	1.35 ± 0.19	1.27 ± 0.15	1.29 ± 0.14	1.40 ± 0.18^*^	1.27 ±0.14 ^#^
**Enzymes (U·L^−1^)**						
SOD	15.4 ± 0.7	15.1 ± 0.9	16.4 ± 0.8 ^*#^	15.6 ± 1.8	16.2 ± 1.5	16.0 ± 1.5
GPx	986 ± 78	953 ± 73	935 ± 84	846 ± 55	849 ± 76	839 ± 64
GR	58.1 ± 6.8	59.7 ± 5.7	60.0 ± 7.3	59.2 ± 5.2	56.7 ± 4.6	59.6 ± 5.0
**Glutathione (nmol·mL^−1^)**						
TGSH	4.85 ± 0.85	6.05 ± 1.42	8.85 ± 1.25 ^***##^	5.78 ± 0.99	5.06 ± 1.01	8.59 ± 1.25 ^***##^
GSH	4.60 ± 0.44	5.53 ± 1.59	8.11 ± 1.44 ^***##^	5.42 ± 0.28	5.09 ± 1.06	8.26 ± 1.25 ^***###^
GSSG	0.305 ± 1.21	0.200 ± 0.092	0.453 ± 0.12 ^**^	0.385 ± 0.152	0.116 ± 0.05 ^**^	0.318 ± 0.181 ^#^

Statistically significant differences shown as asterisk (*) (vs. S1) and hags tag (^#^) (vs. S2): one symbol (*p* < 0.05), two symbols (*p* < 0.01), and three symbols (*p* < 0.001). Cohen’s d values were calculated in the comparisons between two groups in the most representative parameters. TAS: MICT: d = 0.10 (S1 vs. S2), d = 0.07 (S2 vs. S3); HIIT: d = 0.15 (S1 vs. S2), d = 0.12 (S2 vs. S3). SOD: MICT: d = 1.12 (S1 vs. S2), d = 0.95 (S2 vs. S3); HIIT: d = 0.23 (S1 vs. S2), d = 0.25 (S2 vs. S3). TGSH: MICT: d = 1.72 (S1 vs. S2), d = 2.13 (S2 vs. S3); HIIT: d = 1.45 (S1 vs. S2), d = 1.03 (S2 vs. S3). Values are means ± SD. Abbreviations: GPx, glutathione peroxidase; GR, glutathione reductase; GSH, reduced glutathione; GSSG, oxidized glutathione; SOD, superoxide dismutase; TAS, total antioxidant status; TGSH, total glutathione.

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
