# Peer review of "Benefits on Hematological and Biochemical Parameters of a High-Intensity Interval Training Program for a Half-Marathon in Recreational Middle-Aged Women Runners"

_ijerph, 2022, doi:10.3390/ijerph19010498_

Round 1

Reviewer 1 Report

The manuscript was designed to analyze the long-term responses to 12-week moderate-intensity continuous training (MICT) or high-intensity intermittent training (HIIT) programs in premenopausal amateur half-marathon female runners.

The approach is interesting; however, no arguments support the rationale of the training modes (MICT). Why would one type of training influence performance differently than another? The rationale of testing master (older) premenopausal women is also lacking. I understand that age of participants is approximately the same, but aren’t there any academic arguments related to aging?

Please, provide units for BMI data (line 94).

How experienced were your participants?

One thing that concerns the reviewer is the control for the menstrual cycle, which may impact several parameters (especially the derivatives from blood).

The data set required several comparisons derived from the same sample. Wouldn’t a correction be indicated as several ANOVAS were applied?

Interestingly, the HIIT group did not perform a full half-marathon distance during their training period (of course, I am deducting this from the information provided in the manuscript). It would be relevant to inform the number of training sessions in which a whole race (half marathon) was performed. I am asking this because some participants may have to exceed their training routines to accomplish the race. It would partially explain why muscle damage, inflammation, and oxidative stress were more remarkable when performed in a race (S3).

Please, add the number of participants in each group in Tables 1 and 2 for each measurement instant. A Consort figure would be welcome.

Please, explain why the data were collected using different intervals in S2 and S3. Differences of 24 h between S2 and S3 must be discussed in detail.

Consider revisiting titles of the tables, as they must. Refer to columns and rows.

The authors mentioned that participants were recreational but did not provide a performance reference to characterize the sample. Training programs were effective in improving performance, and to what extent? Please, detail.

Speculating is fine as long as plausible mechanisms are also presented. For example, on line 337, you raised a speculation, but no academic arguments were provided to support such an assumption.

As I mentioned before, your comparisons disregard interval differences in the tests (24 vs. 48h).

P values must be shown.                                                   

Finally, I would be pleased to see a regression analysis using the performance of the race (S3) as a dependent variable and all other markers as independent variables. It would allow authors to understand what factors and the magnitude of the changes are more relevant.

Author Response

Dear reviewer, we would like to thank you for accepting reviewing our work and for all the valuable and precise comments and suggestions. You will find changes made in the manuscript, tables and figures highlighted using the track changes mode in MS Word. A point by point response follows after your comments.

The approach is interesting; however, no arguments support the rationale of the training modes (MICT). Why would one type of training influence performance differently than another? The rationale of testing master (older) premenopausal women is also lacking. I understand that age of participants is approximately the same, but aren’t there any academic arguments related to aging?

The rationale to compare haematological parameters and plasma markers between MICT and HIIT training programs in the age group considered in the article has been added at the end of the third paragraph of the Introduction.

Please, provide units for BMI data (line 94).

The units (kg·m-2) have been provided:

How experienced were your participants?

In section 2.1 (Participants) it is indicated that the women who volunteered participate in the study were recruited from different amateur (this word has been newly included) running clubs. They were regular runners with minimum training routines of 5 h over 3 days/week and they had previous experience in running half-marathons. We have also added the mean performance parameters (VO2max, basal heart rate and best half-marathon finishing time).

One thing that concerns the reviewer is the control for the menstrual cycle, which may impact several parameters (especially the derivatives from blood).

A paragraph addressing the reviewer’s concern on the control for the menstrual cycle has been added at the end of 2.1 section.

Interestingly, the HIIT group did not perform a full half-marathon distance during their training period (of course, I am deducting this from the information provided in the manuscript). It would be relevant to inform the number of training sessions in which a whole race (half marathon) was performed. I am asking this because some participants may have to exceed their training routines to accomplish the race. It would partially explain why muscle damage, inflammation, and oxidative stress were more remarkable when performed in a race (S3).

We have added the required information in section 2.2 and the written several lines in section 4.4 as a possible explanation for the differences found at S3 between MICT and HIIT, following the reviewer’s idea and suggestion.

Please, add the number of participants in each group in Tables 1 and 2 for each measurement instant. A Consort figure would be welcome.

The number of participants for each training group has been added in Tables 1, 2 and 3 in the MICT and HIIT column headings. It is explained in 2.3 section that blood samples from all the participants were collected at three different times: S1 (1 week before training began), S2 (48 h before the half-marathon), and S3 (24 h after finishing the race). We did not consider including a consort figure of the experimental design the number of participants (blood samples) for each S1, S2 and S3 group was always n=10. If one sample was missed (for example S3), the complete blood data from the participant were not considered (in the example, S1 and S2 would have not been included in the analysis).

Please, explain why the data were collected using different intervals in S2 and S3. Differences of 24 h between S2 and S3 must be discussed in detail.

The rationale of collecting blood samples in S1, S2 and S3 has been added in section 2.3. Regarding the differences between S2 and S3, in our opinion they have extensively been commented and discussed throughout the subsections of the Discussion. We would appreciate to know which details the reviewer considers on the findings reported in S2 and S3 must be discussed.

Consider revisiting titles of the tables, as they must. Refer to columns and rows.

The titles of the Tables 1, 2 and 3 have been revisited and the columns and rows referred, following reviewer’s suggestion.

The authors mentioned that participants were recreational but did not provide a performance reference to characterize the sample. Training programs were effective in improving performance, and to what extent? Please, detail.

Mean performance parameters (VO2max, basal heart rate and best half-marathon finishing time) from the participants are given in section 2.1.

It was not among the objectives of this work to assess the effectiveness of the training programs in improving the performance, since this was already done in a previous paper [18]. However, at the reviewer’s requirement, at the beginning of the Discussion we have included the mean finishing times reductions obtained after both training programs and a reference to our previous study.

Speculating is fine as long as plausible mechanisms are also presented. For example, on line 337, you raised a speculation, but no academic arguments were provided to support such an assumption.

The reviewer is right. We cannot provide academic arguments to support the assumption (“We can only speculate that, because the HIIT group was exposed to a greater peripheral impact due to their repeated exposure to intense activities, they may have developed adaptive mechanisms regarding tissue filtration that allowed the high intensity effort to have no impact on the hematocrit”). When writing this speculation, we were trying to propose a plausible explanation for our finding. Our experience in previous reviewing processes for other papers showed that it is not unusual that some reviewers or editors ask for hypothesis or speculations trying to explain unprecedented reported findings. In this case, given the reviewer’s reluctance, we decided to delete our speculation in the new manuscript version.

As I mentioned before, your comparisons disregard interval differences in the tests (24 vs. 48h).

We are sorry, but we do not understand the reviewer’s comment. The differences between S2 and S3 sampling points are commented and discussed throughout all the subsections of the Discussion. Please, note that S2 states for a sampling point 48 h before the race (showing the effect of the different training programs) and S3 states for a sampling point 24 h after the half-marathon (showing the impact of the race). In the Discussion, S2 is usually referred as “after the training program” and S3 as “after the marathon race”.

P values must be shown

In section 4.3, P values have been added in the text to support percentage variations

Finally, I would be pleased to see a regression analysis using the performance of the race (S3) as a dependent variable and all other markers as independent variables. It would allow authors to understand what factors and the magnitude of the changes are more relevant.

We have analysed the linear correlations as requested by the reviewer for muscle damage, inflammation and most relevant oxidative stress parameters. A new Table (now number 1) has been added with the results of these analyses showing linear correlation coefficients R2 and P values for the comparisons. We have referred this in Statistical analysis section (2.5), described the results of the table in the first paragraph of the Results section, and discussed in the first paragraph of section 4.4.

Reviewer 2 Report

Dear Authors 

The paper entitled "Benefits on Hematological and Biochemical Parameters of An Intermittent Strength-training Regimen for A Half-marathon in Recreational Middle-aged Women Runners". aims to describe the hematological and biochemical differences of two types of training.
There are some critical issues.
The main one appears to be the statistical analysis which does not seem oriented to the purpose of the study.

Some comments below, excluding the discussion section that probably change with a different results of the study.

  • TITLE: what the Authors mean by "Intermittent Strength-training"? The training program described in method section not appear in line with the term used in title.
  • The introduction section describe mainly the outcomes of this study, but not introduce the meaning of the intervention that the Authors performed on the runners: a different training program between the two groups of the study.
  • The Authors are sure that "regular runners with minimum training routines of 5 h over 3 days/week" can show a BMI=23±3 and therefore some of these cases be in an overweight condition?
  • The aim of the study was the comparison between two different training program: the reviewer suggest to describe both training program in methods section in order to give a better readibiblity and understandability of manuscript.
  • Is not clear the reason why the Authors add eccentric training in HIIT program. In addition, at line 111 were describe downhill running and at line 113 were described uphill and fast running. Please clarify.
  • The session 3 of HIIT training is not clear: "a session that alternated uphill and fast running with a circuit 12 stations of bodyweight resistance exercises performed at maximum intensity."
  • no statistical analysis were perform in order to a between groups comparison. Only in group analysis were compare performed.
  • Please add any differences between groups at baseline for each variables. 
  • In the reviewer opinion, the statistical significant with a 2 group with 10 subjects in not sufficient describe the p value in order to describe the significant. It's advisable to give the power with ES Effect Size analysis. Insert some p value and some not is not scientifically sound (e.s.: 1.  marginal in the HIIT group (P = 0.081); 2. P = 0.092 for sodium and P < 0.05 for chloride)).
  • In the reviewer point of view, it's asvisable a statistical analysis performed on the comparison between groups of the delta value in order to evaluate the magnitude of the changes.
  • In order to provide a more standardization and interpretation of the results is necessary to add the performance during the half marathon: some results in S3 could be conditioned by the time that each subject took during the competition. This aspect must be considered in the follow discussion section.

Author Response

Dear reviewer, we would like to thank you for accepting reviewing our work and for all the valuable and precise comments and suggestions. You will find changes made in the manuscript, tables and figures highlighted using the track changes mode in MS Word. A point by point response follows after your comments.

TITLE: what the Authors mean by "Intermittent Strength-training"? The training program described in method section not appear in line with the term used in title.

We have replaced in the title “intermittent strength-training regimen” for “high-intensity interval training program”.

The introduction section describe mainly the outcomes of this study, but not introduce the meaning of the intervention that the Authors performed on the runners: a different training program between the two groups of the study.

The rationale to compare haematological parameters and plasma markers between MICT and HIIT training programs in the age group considered in the article has been added at the end of the third paragraph of the Introduction.

The Authors are sure that "regular runners with minimum training routines of 5 h over 3 days/week" can show a BMI=23±3 and therefore some of these cases be in an overweight condition?

Our data indicate that the women participating in this study had a mean BMI of 22.5±2.8 kg·m-2 (this figure, with one decimal place, has been corrected in the new version). We agree with the reviewer that, if the overweight threshold is considered strictly from BMI (estimated at 25), some women could have been at the edge of being considered “overweighed”. However, we have data on the percentage of body fat (24.9±3.9) which is in the range of healthy women (23-27). Other data to take into consideration is that the training begun after a holiday’s period (beginning of Section 2.2), which could explain that some participants could have temporarily increased their weight.

The aim of the study was the comparison between two different training program: the reviewer suggest to describe both training program in methods section in order to give a better readibiblity and understandability of manuscript.

In Section 2.2 it is explained “the details of each training program are published” in [18]. Each detailed training program occupies one full page (in Table form) and reproducing both tables in the present paper could fall in a copyright conflict. For this reason, we decided to describe a brief summary of both training programs. It was not the objective of the paper to present a new HIIT program and compare to a validated MICT (this was already done in [18]).

Is not clear the reason why the Authors add eccentric training in HIIT program. In addition, at line 111 were describe downhill running and at line 113 were described uphill and fast running. Please clarify. The session 3 of HIIT training is not clear: "a session that alternated uphill and fast running with a circuit 12 stations of bodyweight resistance exercises performed at maximum intensity."

We used eccentric loading via jump training, downhill running during recovering from uphill running and incorporated training with bodyweight resistance exercises to add a component of strength training to the HIIT program. We have re-written in section 2.2some parts of the text in order to clarify this issue.

no statistical analysis were perform in order to a between groups comparison. Only in group analysis were compare performed.

Yes, it is correct. The objective of the study was to compare the effect of training and the impact of a half-marathon run on hematology and several plasma markers within a MICT and a HIIT program. For this reason, no between-group comparisons were performed. This has been added in the second sentence of the beginning of the Results section.

Please add any differences between groups at baseline for each variables.

This sentence has been added at the beginning of the Results section: “There were no significant statistical differences between groups at baseline (S1) in any of the variables studied.”

In the reviewer opinion, the statistical significant with a 2 group with 10 subjects in not sufficient describe the p value in order to describe the significant. It's advisable to give the power with ES Effect Size analysis.

Cohen’s d values were calculated in the comparisons between S3 vs S2 and S3 vs S1 for muscle damage, inflammation and relevant oxidative stress markers. Their values are given in the figure and table legends with ANOVA P-values. In section 2.5 it has been included a sentence referring to this.

Insert some p value and some not is not scientifically sound (e.s.: 1.  marginal in the HIIT group (P = 0.081); 2. P = 0.092 for sodium and P < 0.05 for chloride)).

We have given the marginal values just to show that, although there were not “statistically significant” differences, there is a slight difference (marginal) which could be relevant when the sample size is small, as is the case.

In order to provide a more standardization and interpretation of the results is necessary to add the performance during the half marathon: some results in S3 could be conditioned by the time that each subject took during the competition. This aspect must be considered in the follow discussion section.

To address this matter, we have performed a regression analysis using the finishing times of the race as dependent variable and the parameters of muscle damage, inflammation and most relevant oxidative stress as independent variables. A new Table (now number 1) has been added with the results of these analyses showing linear correlation coefficients R2 and P values for the comparisons. We have referred this in Statistical analysis section (2.5), described the results of the table in the first paragraph of the Results section, and discussed in the first paragraph of section 4.4.

Reviewer 3 Report

I read with interest the manuscript titled “Benefits on Hematological and Biochemical Parameters of An Intermittent Strength-training Regimen for A Half-marathon in Recreational Middle-aged Women Runners”. The purpose of this study was to investigate to the effects of two half-marathon training programs on hematological and biochemical markers in middle-aged female recreational runners. The findings show that HIIT led to better adaptation to competition maybe because of the repeated exposure to higher oxygen consumption and eccentric loading exercise. The topic is timely and it would provide potential contribution to literature. Overall, the manuscript is well written. I have some concerns as follows.

  1. It would better to describe the rationale for the blood markers including hematological indicators, muscle damage, inflammation and oxidative stress.
  2. The authors need to explain why and how the sample size was determined in this study. Considering the small sample size, the effect sizes (e.g.  Cohen's d value) of the changes in variables might be reported to interpret the findings.
  3. Are there any requirements (i.e. time, fasting, diet, free of coffee) for blood sampling?
  4. Please add the values for the coefficients of variation (CVs) and the reliability of the measures of blood parameters.

Author Response

Dear reviewer, we would like to thank you for accepting reviewing our work and for all the valuable and precise comments and suggestions. You will find changes made in the manuscript, tables and figures highlighted using the track changes mode in MS Word. A point by point response follows after your comments.

It would better to describe the rationale for the blood markers including hematological indicators, muscle damage, inflammation and oxidative stress.

A paragraph explaining the rationale for measuring blood hematological indicators and markers of muscle damage, inflammation and oxidative stress has been added at the end of the Introduction.

The authors need to explain why and how the sample size was determined in this study.

This has been addressed at the beginning of the section 2.1.

Considering the small sample size, the effect sizes (e.g.  Cohen's d value) of the changes in variables might be reported to interpret the findings.

Cohen’s d values were calculated in the comparisons between S3 vs S2 and S3 vs S1 for muscle damage, inflammation and relevant oxidative stress markers. Their values are given in the figure and table legends with ANOVA P-values. In section 2.5 it has been included a sentence referring to this.

Are there any requirements (i.e. time, fasting, diet, free of coffee) for blood sampling?

The requirements asked to the participants before being blood sampled have been included in section 2.3.

Please add the values for the coefficients of variation (CVs) and the reliability of the measures of blood parameters.

Figures are in box-plot format because we think that these graphs are the ones that describe with more detail the information about the distribution of the values. They show the maximum, minimum, median, average and quartile values. We think that including the percentages of CV in these plots will add confusion and the reader could find difficult to follow. In the tables, we used the SD as the dispersion parameter because it is the most used and because ANOVA considers the variance as dispersion parameter. Adding the CV to each parameter in the tables will enlarge considerably these tables (there are tables displaying up to 60 figures which will mean to add 60 more numbers to the existing 120). We politely request to the reviewer to reconsiderate his/her suggestion.

Round 2

Reviewer 1 Report

I thank the authors for their revision. The manuscript has improved in several aspects; however, some parts still deserve attention. For instance, I was expecting authors to explain the rationale of using master recreational runners as they would have different inflammatory responses compared to the training programs, especially regarding the HIIT. In addition, premenopausal changes may also have played a role. The operational aspects forwarded are far from an academic argument.

Minuzzi LG, Rama L, Bishop NC, Rosado F, Martinho A, Paiva A, Teixeira AM. Lifelong training improves anti-inflammatory environment and maintains the number of regulatory T cells in masters athletes. Eur J Appl Physiol. 2017 Jun;117(6):1131-1140. doi: 10.1007/s00421-017-3600-6. Epub 2017 Apr 8. PMID: 28391394.

Sellami,M.;Al-muraikhy, S.; Al-Jaber, H.; Al-Amri, H.; Al-Mansoori, L.; Mazloum, N.A.; Donati, F.; Botre, F.; Elrayess, M.A. Age and Sport Intensity-Dependent Changes in Cytokines and Telomere Length in Elite Athletes. Antioxidants 2021,10,1035. https://doi.org/ 10.3390/antiox10071035

Please, note that I, and I suspect any other reviewers are reluctant to accept speculation. It is fine! However, it must be accompanied by sound academic arguments to be acceptable. This was not the case.

Linear correlations are not linear regressions. Linear regression must be performed with several independent variables or explanatory factors. The data is poorly described as such analyses are also padded with several other information such as Beta, OR, CI, and P values. What type of R2 squares are provided in table 1? Several aspects are also missing. Multicollinearity must be controlled. I offer the authors some clues in which most required parameters are described.

https://www.youtube.com/watch?v=ueNrP5TyZaE

https://www.youtube.com/watch?v=U2p16pCHW3c

There is some text in between Figure 1 and its’ captions (Lines 243 and 245). However, I am not sure whether it is related to the authors.

I am providing an example of the title to help the authors.

Table 2 – Blood parameters of the high-volume and low-intensity (MICT) and low-volume and high-intensity training (HIIT) groups measured 1 week before training (S1), 48 h before the half-marathon(S2), and 24 h after the race (S3).

Figure legends and captions are mixed. One thing is the title; another thing is the caption.

Please reconsider changing the word “insult” in the discussion and throughout the manuscript.

I am unsure what regression analysis was performed, but the discussion is somewhat confusing and mentions correlation coefficients, which are entirely different. A proper analysis may entitle the authors to discuss the role of each parameter on performance. It would add very much to the discussion. Please, revise.

Finally, I assume the authors must acknowledge the limitations of the study.

Author Response

Dear reviewer,

We appreciate all the comments and suggestions. We have incorporated them in the new version of the manuscript. A response to the multiple linear regression subject is also included below in the point-by-point response to your comments and concerns that follows.

I was expecting authors to explain the rationale of using master recreational runners as they would have different inflammatory responses compared to the training programs, especially regarding the HIIT. In addition, premenopausal changes may also have played a role. The operational aspects forwarded are far from an academic argument.

Following the reviewer suggestion, an explanation on the interest in analysing inflammatory, antioxidant and muscle damage markers in masters runners after following a high intensity training program has been added at the end of the Introduction. The references by Sellami et al. (2021) [19] and Minuzzi et al. (2017) [20] have been included to justify an additional rationale of studying this population group. As a consequence, reference numbers from [19] have been changed and conveniently modified throughout the manuscript and in the final list or references.

Please, note that I, and I suspect any other reviewers are reluctant to accept speculation. It is fine! However, it must be accompanied by sound academic arguments to be acceptable. This was not the case.

The reviewer is right and we appreciate his/her comment. For that reason the speculation which was not accompanied by academic arguments has been removed from the original text.

Linear correlations are not linear regressions. Linear regression must be performed with several independent variables or explanatory factors. The data is poorly described as such analyses are also padded with several other information such as Beta, OR, CI, and P values. What type of R2 squares are provided in table 1? Several aspects are also missing. Multicollinearity must be controlled. I offer the authors some clues in which most required parameters are described.

https://www.youtube.com/watch?v=ueNrP5TyZaE. https://www.youtube.com/watch?v=U2p16pCHW3c

I am unsure what regression analysis was performed, but the discussion is somewhat confusing and mentions correlation coefficients, which are entirely different. A proper analysis may entitle the authors to discuss the role of each parameter on performance. It would add very much to the discussion. Please, revise.

We thank the reviewer for the examples provided and the indications on linear correlations and regressions. We considered (as was suggested in the first revision) to run multiple regression analysis using the performance of the race (S3) as a dependent variable and all other markers as independent variables. This multiple linear regression must assume minimum sample sizes of n=20 for each independent variable. Thus, if we would like to perform a multiple regression analysis for 3 variables, we would need n=60 (as is explained in the videos). This sample size far exceeds the sample size we have (n=10) and, for that reason, we decided to run single linear correlations for muscle damage, inflammation and oxidative stress markers. We apologize for the inaccuracies in statistical terms used in the manuscript. In the new version of the manuscript, we have used the term “linear correlation” instead of “linear regression” and include the Pearson product-moment correlation coefficient (r) instead of R2 in Table 1.

There is some text in between Figure 1 and its’ captions (Lines 243 and 245). However, I am not sure whether it is related to the authors.

We have re-arranged the position of Figure 1 in the text. In any case, this and other eventual changes in Figure or Table positions are left for the final layout decided by the editorial board.

I am providing an example of the title to help the authors. Table 2 – Blood parameters of the high-volume and low-intensity (MICT) and low-volume and high-intensity training (HIIT) groups measured 1 week before training (S1), 48 h before the half-marathon(S2), and 24 h after the race (S3).

Tables 2, 3 and 4 have been given new titles according to the example provided by the reviewer.

Figure legends and captions are mixed. One thing is the title; another thing is the caption.

We have followed for Figure editing the editorial template provided in MDPI web site. We have included a title in the first line of Figure 1 to be consistent with Figures 2 and 3. We would like to express our willingness to change the final layout according to the reviewer’s suggestion including Figures titles as headings and we leave the final Figure title and legend layout at the editorial discretion.

Please reconsider changing the word “insult” in the discussion and throughout the manuscript.

The word “insult” has been replaced with other synonym expressions throughout the text: page 2 (Introduction) and pages 11 and 13 (Discussion).

Finally, I assume the authors must acknowledge the limitations of the study.

A final paragraph in the Discussion section has been added acknowledging the limitations of the study.